



# Generalized drought index: A novel multi-scale daily approach for drought assessment

João António Martins Careto[1], Rita Margarida Cardoso[1], Ana Russo[1], Daniela Catarina André Lima[1], Pedro Miguel Matos Soares[1]

[1] Universidade de Lisboa, Faculdade de Ciências, Instituto Dom Luiz, 1749-016 Lisbon, Portugal.

*Correspondence to: João Careto (jacareto@ciencias.ulisboa.pt)*

**Abstract.** Drought is a complex climatic phenomenon characterized by water scarcity recognized as the most widespread and insidious natural hazard, posing significant challenges to ecosystems and human society. In this study, we propose a new daily based index for characterizing droughts, which involves standardizing precipitation

and/or precipitation minus potential evapotranspiration data. The performance of this new index is assessed with data from the evaluation runs of the Coordinated Regional Climate Downscaling Experiment for the European domain and the observational data from the Iberian Gridded Dataset, covering the period from 1989 to 2009. Comparative assessments are conducted against the daily Standardized Precipitation Index (SPI), the Standardized Precipitation Evapotranspiration Index (SPEI), and a simple Z-Score standardization of climatic variables. Seven

different accumulation periods are considered (7, 15, 30, 90, 180, 360, and 720 days) with three drought levels: moderate, severe, and extreme. The evaluation focuses mainly on the direct comparison amongst indices, added value assessment using the Distribution Added Value and a simple bias difference for drought characteristics. Results reveal that not only does the new index allow the characterization of flash droughts, but also demonstrates added value when compared to SPI and SPEI, especially for longer accumulation periods. In comparison to the

Z-Score, the new index shows slightly greater gains, particularly for extreme drought events at lower accumulation periods. Furthermore, an assessment of the spatial extent of drought for the 2004-2005 event is performed using the observational dataset. All three indices generally provide similar representations, except for the Z-Score, which exhibits limitations in capturing extreme drought events at lower accumulation periods. Overall, the findings suggest that the new index offers improved performance and adds value comparatively to

similar indices with a daily time step.

## 1.      Introduction

Drought is known to be one of the most impactful and costliest weather-related disasters, affecting ecosystems, the economy, and sectors such as agriculture, health, and water management (Wilhite, 2000; Rhee et al., 2010; Vicente-Serrano et al., 2013; Wang et al., 2014; 2017; Lai et al., 2019). Amongst all natural disasters, droughts

can spread further and have the longest duration (Jain et al., 2010), developing most often on a slow manner, while at the same time, their effects can linger in the environment long after the end of the event (Vicente-Serrano et al., 2012; Hunt et al, 2014).

Over the years, numerous indices have been developed to assess drought conditions, particularly in terms of intensity and duration. One of the first proposed drought indices is the Palmer Drought Severity Index (PDSI,

Palmer, 1965; Alley, 1984), which enables the measurement and evaluation of both wet and dry conditions. The PDSI standardizes the balance between monthly precipitation and atmospheric demand by incorporating potential evapotranspiration in its formulation. While this index was a landmark, it does reveal certain shortcomings. Its performance is enhanced for the region where the index was initially defined with its outputs being heavily influenced by the chosen calibration period. Therefore, PDSI revealed problems related to its spatial comparison



and application. To address some of these issues, Wells et al. (2004) introduced the self-calibrated PDSI, which
allows for spatial comparison and identifies extreme wet and dry events as rare occurrences. However, fixed
timescales for computing the index remain a concern. Further developments were introduced during the following
years to address these caveats. The Standardized Precipitation Index (SPI, McKee et al., 1993) is one of the indices
developed that tackled the comparability and temporal scales issues (Guttman 1998; Hayes et al. 1999). SPI is a

straightforward Standardized index only requiring monthly precipitation, representing it as a standard deviation
from its mean. SPI overcomes the limitations of the self-calibrated PDSI by enabling the computation of the index
at various timescales. Nevertheless, the consideration of just precipitation by SPI could be a limiting factor
depending on the climatic dominating conditions in certain regions. However, with anthropogenic climate change,
rising temperatures and the subsequent increases in evapotranspiration can also significantly increase the impact

of drought events (Hu and Wilson, 2000; Vicente-Serrano et al., 2010). Therefore, not including the influence of
atmospheric evaporative demand in a drought index becomes imperative (Vicente-Serrano et al., 2010; Svoboda
and Fuchs, 2016). To address this need, Vicente-Serrano et al. (2010) proposed the Standardized Precipitation
Evapotranspiration Index (SPEI), which was further developed by Beguería et al. (2014). SPEI combines all the
features and advantages of SPI together with the inclusion of atmospheric evaporative demand represented by the

potential evapotranspiration. Both SPI and SPEI are indices that require data to be fitted to a theoretical Probability
Density Distribution (PDF). In the literature, numerous PDFs have been considered. For SPI, distributions such
as Pearson type III (Vicente-Serrano et al., 2006) or Gamma (Mkee et al., 1993; Edwards et al., 1997; Wang et
al., 2022; Zhang et al., 2023) have been commonly employed. On the other hand, the 3-parameter log-logistic
(Beguería et al., 2014; Wang et al., 2015; Ma et al., 2020) and the Generalized Extreme Value (Stagge et al., 2015;

Wang et al., 2021; Zhang et al., 2023) distributions have been widely used for SPEI. However, the best distribution
to fit the data is still not clear, as the same distribution may perform differently for distinct regions (Stagge et al.,
2015; Monish and Rehana, 2020; Zhang and Li, 2020). For instance, for a global dataset, Stagge et al. (2015)
concluded that the Gamma (Weibull) for long (short) accumulations was the best distribution for SPI, while the
Generalized Extreme value was the best distribution to fit SPEI. On the other hand, Zhang and Li. (2020),

concluded that the Log-Logistic distribution could be used as an alternative when analyzing SPI for a large river
basin in China. At the same time, the Log-logistic distribution is known to be resilient to the presence of outliers
(Ahmad et al., 1988) and for the Iberian Peninsula (Vicente-Serrano et al., 2010; Beguería et al., 2014) was deemed
the best function for fitting the data for SPEI. Usually, the SPI and SPEI indices only rely on a single probability
density distribution, even for large regions. To overcome this issue, there are methods to estimate the underlying

distribution and associated parameters, which could, however, become computationally infeasible for large
datasets (Guttman, 1999). At the same time, the method considered to estimate the parameters of a single
distribution could also be computationally demanding,

Simpler drought indices that do not require fitting to a distribution also exist. One is the Z-Score which is computed
for precipitation or to the difference between precipitation and evapotranspiration by subtracting the long-term

mean and dividing the result by the long-term standard deviation (Komuscu, 1999; Morid et al., 2006; Patel et al.,
2007; Akhtari et al., 2009; Dogan et al., 2012; Jain et al., 2015). Slightly different formulations for this index also



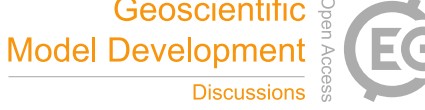

exist such as used by Zhang et al. (2022a) and (2022b), or the China Z-Index (Wu et al., 2001), and is also considered in the by the Reconnaissance Drought Index (Tsakiris and Vangelis, 2004). The advantage of the Z-Score index lies in its simple calculation being considered an alternative to indices which require fitting data to a distribution such as the SPI or SPEI, being capable of accommodating missing values. Similarly, the Z-Score also represents a Standardized departure from the mean. However, the Z-Score may not effectively represent the shorter timescales since precipitation data is skewed (Edwards and McKee, 1997). Additionally, the performance of the index may vary in regions with diverse precipitation or potential evapotranspiration patterns, where data do not assume a normal distribution. This can affect the accuracy and reliability of the index.

Although droughts are in general known to be a slowly evolving phenomenon (Wilhite and Glantz, 1985; Mishra and Singh, 2010), recently the concept of flash drought has emerged (Wang et al., 2021;2022; Zhang et al., 2022a; Christian et al., 2023). These types of extreme events are characterized by a sudden onset, fast aggravation, and end (Christian et al., 2023). Depending on the type of climate, these short-duration events may threaten the water supply and cause significant reductions in crop yield at critical stages of plant development (Meyer et al., 1993; Dai, 2011; Vicente-Serrano et al., 2013; Hunt et al., 2014). Flash droughts can only be identified with daily drought indices, due to their sub-monthly timescale nature. Therefore, concurrently with the widespread observation-based and model-based daily datasets such as the National Gridded Dataset for the Iberia Peninsula (IB01; Herrera et al., 2019), or the World Climate Research Program Coordinated Regional Climate Downscaling Experiment (Giorgi et al., 2009; 2021, Gutowski et al., 2016) for the European domain (EURO-CORDEX, Jacob et al., 2014; 2020), the development of new drought indices with a daily time step has emerged (Wang et al., 2015; 2021; 2022; Jia et al., 2018; Li et al., 2020; Ma et al., 2020; Onuşluel Gül et al., 2021; Zhang et al., 2022a; 2022b; Zhang et al., 2023). At the same time, most of these indices are still fitted to a probability distribution and/or are not standardized. Wang et al. (2015) used a daily version of the SPEI to understand if there has been any improvement in drought conditions. The authors report that the daily SPEI can provide a more comprehensive understanding of drought dynamics at a finer temporal scale. Li et al. (2020) proposed the Standardized Antecedent Precipitation Evapotranspiration Index. The index is first compared against the monthly PDSI and SPEI and soil moisture, revealing a similar performance against SPEI at the monthly scale, while outperforming at the weekly scale. Ma et al. (2020) computed a daily SPEI index and compared it with the traditional monthly version. The authors reported that the daily index can capture more detailed drought events compared to the monthly counterpart. Wan et al. (2023) also considered a daily SPEI index to in order to determine the trend of drought severity and duration over the course of 40 years (1979-2018) for China. The authors also concluded that the potential evapotranspiration was the dominant climatic factor influencing drought for most of the region. Zhang et al. (2022a) proposed the Daily Evapotranspiration Deficit Index and compared the results against the Meteorological Drought Composite Index and SPEI for four drought events which occurred in China. The proposed index was able to better capture the start and end of the events, as well as the peak intensity. Still indices such as the SPI or the SPEI computed at the daily scale also prove to be demanding. The parameter estimation and the subsequent fitting to a theoretical distribution may not only be computationally expensive but also may result in a poor fit. At sub-monthly aggregation scales the presence of outliers could as well hinder the parameter estimation and fit, but





also generate values that might fall outside the range of the chosen distribution. Furthermore, periods with no

precipitation may also pose difficulties in computing the SPI index (Beguería et al., 2014).

Nevertheless, daily indices are still in their early ages in comparison with the diverse monthly drought indices available. Motivated by the shortcomings illustrated before, in the present study, we propose a new daily drought index, the Generalized Drought Index (GDI). The GDI is an index similar to SPI or SPEI in the sense of standardizing data to follow the standard normal distribution, allowing the evaluation of both short and long-time-

scale droughts with a daily time step. Furthermore, GDI allows for a generalized fitting distribution which is empirically based and, thus the index accepts alternative variables for drought assessment and not only precipitation. For instance, actual evapotranspiration could be considered as an alternative to the usual precipitation minus potential evapotranspiration (P-PET). Moreover, the new index may be perceived as an alternative for removing skewness and kurtosis from climate data. Table 1 displays a summary of all indices

presented here.

Here, the GDI index is computed for the Iberian Peninsula region using the IB01 dataset and EURO-CORDEX evaluation simulations. Our study contributes to the ongoing efforts to develop more effective drought monitoring tools and provides a valuable instrument for decision-makers and stakeholders to better manage the impacts of flash-droughts and longer droughts, in a consistent and solid way. Our proposed index can be easily implemented

in regions with limited climatic variables and can help improve the accuracy and reliability of drought assessment, requiring solely long-time series.

The following section introduces the IB01 and EURO-CORDEX data, as well as the methodology for computing the GDI, the SPI and SPEI, and finally the simple Z-Score standardization. Afterwards, the results are presented in section 3, followed by a Discussion and Conclusions in section 4.






**Table1. Examples of drought indices.**

| Index | Reference | Aggregation | | | Variable | | | Method | | |
|---|---|---|---|---|---|---|---|---|---|---|
| | | Monthly | Weekly | Daily | PR | PR-PET | AET | Fit to Dist. | Empirical Fit | Equation |
| ------ | ------ | | | | | | | | | |
| PDSI | Palmer, (1965) | X | | | | X | | | | X |
| sc-PDSI | Wells et al. (2004) | X | | | | X | | | | X |
| SPI | Mckee et al. (1993) Zhang et al. (2023) | X | | X | X | | | X | | |
| SPEI | Vicente-Serrano et al. (2010) Ma et al. (2020) Zhang et al. (2022a) | X | X | X | | X | | X | | |
| Z-Score | Komusou, (1999) | X | X | X | | X | | | | X |
| RDI | Tsakiris and Vangelis, (2004) | X | | | | X | | | | X |
| SAPEI | Li et al. (2020) | X | X | | | X | | X | | |
| DEDI | Zhang et al. (2022a) | | | X | | X | | X | | |
| GDI | ------ | X | X | X | X | X | X | | | X |



## 2.    Data and Methods

### 2.1.    Study area and Data

The Iberian Peninsula exhibits a diverse and complex climate influenced by its geographical position, surrounded by the Atlantic Ocean to the north and west, and the Mediterranean Sea to the south and east. In the northern regions of the Iberian Peninsula, such as Galicia and northern Portugal, a maritime climate prevails, characterized by mild winters and cool summers. The Atlantic Ocean influence brings relatively high precipitation throughout the year (Rios-Entenza et al., 2014). Towards the south, the climate shifts to a more Mediterranean type, with hot and dry summers. Winters remain mild and relatively wet compared to the summer months. The Mediterranean climate is associated with distinct wet and dry seasons, with most of the rainfall occurring during the winter (Peel et al., 2007). Droughts are a recurring and a significant challenge for the Iberian Peninsula. The region has a long history of drought events, with a clear drying trend throughout the 20$^{th}$ century, mainly due to an increase in temperature (Fonseca et al., 2015; Páscoa et al., 2021). Climate change projections suggest that the frequency and intensity of droughts may increase in the future (Soares et al., 2023b). Rising temperatures and changing precipitation patterns may exacerbate water scarcity and put additional stress on the region's ecosystems. (Soares et al., 2017; Cardoso et al., 2019; Carvalho et al., 2021; Soares and Lima 2022).

### 2.1.1.    IB01 Observational Dataset

The IB01 Observational dataset (Herrera et al., 2019) is a high-quality dataset that offers daily values for precipitation, as well as minimum and maximum temperatures, at a spatial resolution of 0.1o. This dataset was constructed using an extensive network of quality-controlled observational weather stations (a maximum of 3486 for precipitation and 275 for temperatures) across the Iberian Peninsula during the period from 1971 to 2015. Herrera et al. (2019) reported that IB01 effectively captures the spatial patterns of the mean and extreme precipitation and temperatures. Regarding the mean and extreme precipitation, the authors demonstrated that IB01 exhibits a more realistic pattern than E-OBS, while for temperature, both datasets demonstrate comparable performance.

The IB01 dataset has been employed in numerous studies to characterize the present climate and used as a benchmark for evaluating the performance of a set of EURO-CORDEX simulations in reproducing the present climate over Iberia (Herrera et al. 2020; Páscoa et al., 2021; Careto et al., 2022a; 2022b; Lima et al., 2023a; 2023b; Soares et al., 2023b). Herrera et al. (2020) evaluated the performance of the EURO-CORDEX over the Iberian Peninsula and characterized the observational uncertainty with the use of the IB01, E-OBS-v19e, and MESAN-0.11 datasets. Páscoa et al. (2021) employed this dataset to assess the recent trends in drought events across Iberia. Careto et al. (2022a) and (2022b) evaluated the added value of using high-resolution simulations from EURO-CORDEX in characterizing means and extremes of precipitation and temperature over the Iberian Peninsula. More recently, Lima et al. (2023a) considered the IB01 dataset as the reference to evaluate the accuracy of a set of historical EURO-CORDEX simulations in representing the main properties of the observed climate within mainland Portugal. Based on this evaluation, a weighted multi-variable multi-model ensemble of EURO-



CORDEX simulations was built and used to characterize both the mean climate, extremes and indices (Lima et al.
2023a and 2023b), as well as water scarcity conditions over Portugal (Soares and Lima, 2022) throughout the 21st
century. Based on the same weighting methodology, Soares et al. (2023b) projected the future of drought events
across the Iberian Peninsula. Finally, IB01 was used to critically assess the CMIPs quality to project the recent
past climate of Iberia (Soares et al, 2023a). In this study, a conservative remapping technique (Schulzweida et al.,
2019) is employed to transform the IB01 dataset into the EURO-CORDEX 0.11o resolution.

### 2.1.2. EURO-CORDEX Evaluation Simulations

The COordinated Regional Downscaling EXperiment (CORDEX) is supported by the World Climate Research
Programme to create an ensemble of high-resolution projections from Regional Climate Models (RCMs). The
main objective was to generate data at user-relevant scales and provide valuable support for research on climate
change impacts and adaptation (Giorgi et al. 2009; Gutowski et al. 2016). Within the CORDEX framework,
EURO-CORDEX (Jacob et al. 2014; 2020) serves as a specific branch dedicated to the European region (Fig. 1),
which fostered runs with horizontal resolutions of 0.44o, 0.22o and 0.11o. These simulations consist of an
evaluation experiment for the 1989-2008 period, forced by the ERA-Interim Reanalysis (Dee et al., 2011) and a
Historical-Scenario experiment where RCMs downscale the Coupled Model Intercomparison Phase 5 (CMIP5)
Global Climate Models (GCMs) for at least the 1971-2100 period.

In this study, only the evaluation simulations are considered since they have daily synchronized climate data,
enabling the assessment of the added value of a set of indices, introduced in the following sub-sections, against
the IB01 dataset. The variables needed were retrieved through the Earth System Grid Federation (ESGF) data
portal, including daily total precipitation and maximum and minimum 2-m daily temperature at 0.11° resolution.
In total 12 EURO-CORDEX ensemble members were considered. Table S1 from the supplementary material
summarizes the regional climate information used in this study and Fig. S1 displays the orography of both the
IB01 and EURO-CORDEX datasets on their respective original resolutions.

Extensive assessments have been conducted on all EURO-CORDEX evaluation simulations, focusing on key
variables such as precipitation and temperatures (Vautard et al., 2013; Kotlarski et al., 2014; Katragkou et al.,
2015; Casanueva et al., 2016a; 2016b; Prein et al., 2016; Soares and Cardoso, 2018; Herrera et al., 2020; Cardoso
and Soares, 2022, Careto et al., 2022a; 2022b; Molina et al., 2023). Herrera et al. (2020) gauged precipitation and
temperature using an ensemble of eight EURO-CORDEX RCMs over the Iberian Peninsula against three
observational datasets. The authors reported a good spatial agreement between models and observations,
particularly for temperature. However, the agreement decreased when considering extreme events, with model
precipitation showing a larger uncertainty compared to temperature. Assessments of the added value of using
high-resolution simulations were conducted by Soares and Cardoso (2018) for the European domain, showing
significant improvements in the representation of precipitation patterns, particularly for extremes. In a recent
study, Cardoso and Soares (2022) evaluated the added value of temperature (maximum and minimum) in Europe,
while in Careto et al. (2022a; 2022b) the added value of precipitation and maximum and minimum temperature is
assessed over Iberia, at higher resolution. In all cases, there is added value in the precipitation downscaling of the





210 EURO-CORDEX 0.11o simulations, whilst this was not the case for temperatures. The authors reported difficulties from the RCMs in representing the snow-albedo-feedback for complex terrain. Moreover, for the evaluation simulations, the assimilation of observed data by the ERA-Interim makes it difficult for the regional models to reveal significant added value. Despite those limitations, the higher resolution simulations for the coastal regions displayed gains comparatively to their respective forcing simulations.

## 2.2. Potential Evapotranspiration

Potential Evapotranspiration (PET) represents the maximum atmospheric water demand and is a requirement for the computation of several drought indices (Vicente-Serrano et al., 2010; Li et al., 2020; Zhang et al., 2022a; 2022b). The FAO-56 Penman-Monteith formula (Allen et al., 1998) is one of the most widely used approaches to calculate PET. Although it was specifically designed for non-stressed grass cover, is considered the most accurate

220 estimate. However, it requires multiple variables, some of which may not be readily available, posing a drawback to its practical implementation. An alternative approach, known for its simplicity, is the Thornthwaite formulation (Thornthwaite, 1948), which only requires latitude and temperature as inputs. However, studies have shown that the Thornthwaite formulation underestimates PET in arid and semiarid regions while overestimating it in humid tropical or equatorial regions (van der Schrier et al., 2011). Therefore, in the context of climate change, this

225 equation is not the best option for computing PET (Beguería et al., 2014).

As a compromise between formulation complexity and data availability, a modified version of the Hargreaves formulation is thus considered in this study (Droogers and Allen, 2002). The Modified Hargreaves is similar to the original Hargreaves method, in which beyond the incorporation of maximum and minimum temperature, the precipitation is also integrated. Precipitation data is commonly accessible in most modeling and observational

230 datasets and can serve as a proxy for cloud cover and humidity. In this study, a daily version of the Modified Hargreaves formula is implemented (Farmer et al., 2011):

$$PET = 0.0019 * 0.408 * RA * (Tavg + 21.0584)(TD - 0.0874 * P)^{0.6278} \tag{1}$$

## 2.3. Drought Indices

### 2.3.1. Standardized Precipitation and Standardized Precipitation Evapotranspiration Indices

In this section, the Standardized Precipitation Index (SPI, McKee et al. 1993) and the Standardized Precipitation

235 Evapotranspiration Index (SPEI, Vicente-Serrano et al. 2010;) are presented. Both SPI and SPEI are commonly used (Edwards, 1997; Vicente-Serrano et al. 2006; 2010; Beguería et al. 2014; Wang et al. 2022; Zhang et al. 2022a; 2022b) with the former being calculated based solely on precipitation (hereafter PR) and the latter on a simplified water balance (Precipitation minus Potential Evapotranspiration, hereafter PR-PET). Probabilistic indices such as these allow for a standardized juxtaposition and comparison across different spatial areas or

240 between climate zones (Vicente-Serrano et al., 2010; Pohl et al., 2023)

To compute either the SPI or the SPEI, first, the PR and PR-PET data must be aggregated into the desired timescale through a moving window with a length equal to the timescale, i.e., a daily value is computed as the sum of the day under analysis (d) and the previous $s - 1$ days where s is the timescale (in days):





$$X_d = \sum_{d-(s-1)}^{d} data \tag{2}$$

Subsequently, a daily yearly mean is obtained from a moving window of 31 days centred on each day d:

$$Se = \frac{1}{31Y} \sum_{i=1}^{Y} \sum_{j=D-15}^{D+15} X_{d_{i,j}} \tag{3}$$

Where Y is the total number of years and D is the day of the year. For instance, 1st January corresponds to day 1 and 31st December to day 366. To ease all computations, all years are considered to have 366 days in order to include the 29th of February from leap years. Consequently, the value for 29th February from non-leap years is considered a missing value. Thus, $Se$ is an annual mean cycle. Thirdly, this annual cycle is removed from the $X_d$ series:

$$X_a = \sum_{i=1}^{Y} \sum_{j=1}^{366} X_{d_{i,j}} - Se_j \tag{4}$$

Traditionally, the removal of the seasonal cycle is not performed for the SPI and SPEI. However, it can be regarded as a step to remove days without precipitation, which is relevant in the case of the SPI index. Usually in those situations, a factor is considered for precipitation data (Stagge et al., 2015; Wang et al., 2022; Zhang et al., 2023).

Afterwards, the $X_a$ series are adjusted to a theoretical distribution. The log-logistic distribution was chosen to fit

$X_a$ for both SPI (Zhang and Li, 2020) and SPEI (Vicente-Serrano et al., 2010; Beguería et al., 2014). Therefore, the difference between the two indices lies solely in the inclusion of PET for SPEI. To avoid issues when fitting the data to the distribution, first the values of the $X_a$ series are shifted to positive values above 0. This change does not affect the distribution or the final value. The log-logistic density distribution has the following expression:

$$f(x) = \frac{\beta}{\alpha} \left( \frac{x-\gamma}{\alpha} \right)^{\beta-1} \left[ 1 + \left( \frac{x-\gamma}{\alpha} \right)^{\beta} \right]^{-2} \tag{5}$$

The three parameters β (shape), α (scale) and γ (location) can be estimated via the maximum likelihood or with Probability Weighted Moments (PWM, Hosking, 1986; 1990). Following Beguería et al. (2014), the unbiased estimator for PWM (Hosking, 1986) was considered:

$$W_s = \frac{1}{N} \sum_{i=1}^{N} \frac{\binom{N-i}{s}}{\binom{N-1}{s}} X_{d_i} = \frac{1}{N} \sum_{i=1}^{N} \frac{\Gamma(N-i+1)/(\Gamma(s+1)\Gamma(N-i-s+1))}{\Gamma(N)/(\Gamma(s+1)\Gamma(N-s))} X_{a_i} \tag{6}$$

Moments $W_s$ of different orders $s$ can be computed easily via software programming tools. Γ denotes the gamma function for Natural numbers including 0. From the first three moments ($W_0$, $W_1$ and $W_2$) it is possible to obtain

the three parameters for the Log-Logistic (Singh et al., 1993):

$$\beta = \frac{2W_1 - W_0}{6W_1 - W_0 - 6W_2} \tag{7}$$





$$\alpha = \frac{(W_0 - 2W_1)\beta}{\Gamma\left(1 + \frac{1}{\beta}\right)\Gamma\left(1 - \frac{1}{\beta}\right)} \tag{8}$$

$$\gamma = W_0 - \alpha\Gamma\left(1 + \frac{1}{\beta}\right)\Gamma\left(1 - \frac{1}{\beta}\right) \tag{9}$$

To convert the $X_a$ series into SPI or SPEI, the Cumulative Distribution Function (CDF) of the Log-Logistic is required to obtain the accumulated probability:

$$F(x) = \left[1 + \left(\frac{\alpha}{X_a - \gamma}\right)^\beta\right]^{-1} \tag{10}$$

Having the accumulated probabilities, the indices can be easily obtained following the classical approximation of Abramowitz and Stegun. (1965):

$$P = 1 - F(x) \tag{11}$$

$$P = 1 - P, if\ P > 0.5 \tag{12}$$

If P is above 0.5, then the signal of the final index is also reversed.

$$W = \sqrt{-2\ln(P)} \tag{13}$$

$$SPEI = W - \frac{C_0 + C_1 W + C_2 W^2}{1 + D_1 W + D_2 W^2 + D_3 W^3} \tag{14}$$

With $C_0 = 2.515517, C_1 = 0.802853, C_2 = 0.010328, D_1 = 1.432788, D_2 = 0.189269\ and\ D_3 = 0.001308$.

### 2.3.2. Z-Score Index

Z-Score method is a straightforward approach used to standardize a dataset based on its mean and standard deviation (Komuscu, 1999; Patel et al., 2007; Akhtari et al., 2009; Jain et al., 2015). It follows a simple rationale:
1) obtain the accumulated series and remove its seasonal cycle, as described in section 2.3.1; 2) remove the mean and divide the result by the standard deviation to get the $X_a$ anomalies. This ensures that all data points have the same statistics for mean and standard deviation. However, it is important to note that while the mean and standard deviation will be consistent across all points, the underlying distribution and its parameters describing the data at each location may vary. Still, for long accumulations and as a consequence of the central limit theorem, the Z-
Score and the standardized indices approach each other. The Z-Score can be computed by:

$$Z - Score = \frac{X_a - \overline{X_a}}{\sigma(X_a)} \tag{15}$$

### 2.3.3. Generalized Drought Index

A new index, the Generalized Drought Index (GDI) is proposed here as an alternative to the commonly used standardized drought indices, such as the SPI or the SPEI, both described in section 2.3.1. The GDI is also a
standardized index but introduces three upgrades which are particularly interesting when addressing drought impacts which often occur at sub-monthly scales:

- It can be calculated using any daily aggregation. For instance, the 7-, 15-, 30-, 90-, 180-, 360-, and 720-days



were chosen, ranging from weekly to biannual aggregations. Regardless of the timescale chosen, a daily index is obtained, allowing an assessment of flash droughts, which were not possible with monthly indices.

● Since fitting to a distribution is not required, any variable relevant to drought characterization can be considered as input, such as PR or PR-PET, actual evapotranspiration or PR divided by PET.

● Relies on a unique spline adjustment technique to smooth the cumulative histogram. The main advantage is the automatic fit of the empirical distribution to the data for different sites, resulting in an enhanced index.

Figure S2 (in supplemental material) features a distribution from the individual grid points of the coefficient of
determination, comparing the empirical cumulative distribution against the log-logistic distribution and the spline adjustment for both PR and PR-PET and considering all accumulation. This figure clearly reveals that the spline adjustment outperforms the theoretical log logistic fit.

To compute the GDI, the $X_a$ series anomalies obtained in the subsection 2.3.1 are considered. The next step is to compute a histogram of the data. The Freedman-Diaconis rule is used, which gives an optimized estimate for the
bin width based on the data variability and length:

$$inc = 2 * \frac{IQR}{\sqrt[3]{N}} \tag{16}$$

Where IQR is the interquartile range and N the length of the $X_a$ series. The histogram is defined between the minimum and maximum values and is tailored specifically for each time series. Following Soares and Cardoso, (2018) the histogram series are normalized by the sum of all bins:

$$X_d = \frac{hist(X_d)}{\text{sum}\big(hist(X_d)\big)} * (1 - N^{-1}) \tag{17}$$

Subsequently, a cumulative sum of each bin is considered. At this stage, the bins of the cumulative histogram
were treated as data (x, y) points, where x represents the endpoint between the bin edges and y represents the corresponding probability. A value proportional to the length of data ($1/N$) is appended at the minimum edge of the first bin, corresponding to the minimum value of the $x_d$ series. It is important to avoid 0s and 1s, since the cumulative distribution of the normal distribution tends to infinity for a probability of 0 and 1. Therefore the factor $(1 - N^{-1})$ was considered, slightly scaling down the value of all bins. Afterwards, a cubic spline technique
(Fritsch and Butland, 1984) is used to smooth the cumulative histogram. This approach allows to obtain the probability of a certain value to occur, without fitting to a theoretical distribution.

This method allows estimating intermediate probabilities between the cumulative histogram points, resulting in a continuous and smooth representation of the underlying distribution, bounded by the probability of the minimum value and the last bin, allowing the preservation of the daily time step for the final index. Afterwards, the original
$X_a$ values are converted into accumulated probabilities using the interpolated accumulated histogram. By using the inverse of the normal distribution, one can transform these probability values into a standardized series following the standard normal distribution with a mean of 0 and a standard deviation of 1. It is important to note that the feasibility of this approach depends on the length of the original time series, as the statistics from longer time series will tend to align more closely with the parameters of the normal distribution. This is similar to what
occurs for both SPI and SPEI (McKee et al., 1993; Pohl et al., 2023). Figure 1 introduces a flowchart to guide the





users on the steps needed to obtain the GDI.

### 2.4.    GDI evaluation

The performance of GDI, SPI, SPEI, and Z-Score is assessed using the IB01 dataset for each single location to generate quantile-quantile plots, which allow to determine the underlying distribution of all-time series relative

to the standard normal distribution. The percentiles considered for evaluation are constituted by a sequence from the 10th to the 90th percentile, with an increment of 10. With this inter-comparative analysis one can inspect the underlying distribution of the data and how close it is to the theoretical standard normal distribution. A wider vertical spread represents deviations of the time series from normality (linear line). The conforming of results to the standard normal distribution is of paramount importance in various statistical analyses, as it facilitates

meaningful comparisons and allows for the application of well-established statistical techniques. When data closely follows the standard normal distribution, it exhibits known statistical properties, simplifying the interpretation of the results (e.g., equal mean and median, 68% of the data falls within one standard deviation of the mean, 95% of the data falls within two standard deviations of the mean). In the context of drought indices, compliance with normality assumptions is crucial for accurately characterizing drought severity and frequency.

Moreover, the standard normal distribution allows for direct comparison across different spatial areas and time periods, which is of particularly relevance for assessing drought severity and patterns on both regional and global scales (Guttman 1998; Hayes et al., 1999; Vicente-Serrano et al., 2006; 201; Beguería et al., 2014). As a complement, statistics including the mean, median, standard deviation, interquartile range, skewness, Yule-Kendall skewness, and kurtosis were computed for all indices, from all land points of observations. Finally, to

further investigate the differences between the GDI and the other indices, a Pearson correlation and the Root-Mean-Squared-Error (RMSE) are also computed. All the metrics mentioned to this point had the main goal of assessing the overall performance for daily time series.

Regarding the drought characteristics, several aspects are examined such as the mean event severity, decadal frequency, mean event duration and daily spatial drought extent. The drought levels considered are as follows:

moderate drought with the index below -0.5, severe drought with index below -1 and extreme drought with index below -1.5 (McKee et al., 1993; Soares et al., 2023b). The drought frequency can be defined as the average number of times any index falls below the specified threshold per decade. Mean event severity is computed by dividing the sum of all days with the index below the defined thresholds, against the total number of events. The mean event duration is computed in a similar way and is defined as the division of the total number of days by the total

number of drought events. All the statistics were computed for all indices and for the 7-, 15-, 30-, 90-, 180-, 360- and 720 timescales considering all models and observations for each grid point. A comparison is then performed with the IB01 dataset by considering a spatial correlation and root mean squared error between the GDI and other indices, for moderate drought only.



**Figure 1. Flowchart for the construction of the Generalized Drought Index.**

An analysis of the spatial extent of drought classes (moderate, severe, and extreme) is performed for the IB01 dataset for a case study. The spatial extent is computed for each day and is defined as a percentage of land points

with any index below the given thresholds. In this assessment, only the results for the observations are assessed, even though the EURO-CORDEX models are synchronized among themselves and with observations, differences would still arise, hindering a correct assessment of the event. This analysis is performed for the extreme drought





that affected the Iberia Peninsula in 2004 and 2005, where the evaluation period corresponds to two hydrological years, starting in October 2004 and ending in September 2006. Although the Z-Score is also a standardization, from this point forward the term standardized indices will be used to describe the SPI, SPEI and GDI, in which the results comply to the standard normal distribution. Moreover, the spatial average time series of all drought indices for all time aggregations were also analysed.

A Distribution Added Value (DAV, Soares and Cardoso, 2018) assessment is also performed. The DAV allows a comparison of the similarity between model and observational distributions across two different indices. This assessment is performed for the GDI against the SPI or SPEI and also the GDI against the Z-Score for each grid point. To compute the DAV, a histogram is first constructed. For GDI, SPI and SPEI the limits considered are -5 to 5, while for the Z-Score the limits are wider, ranging from -15 to 15. The bin width was set to be constant for all datasets and is determined by the Freedman-Diaconis rule described earlier. In this context, the 75th and 25th percentiles are taken from a theoretical standard normal distribution. Afterwards, each histogram is normalized by dividing each bin with the sum of all bins. From the histograms a Perkins Skill Score ($S_{index}$; Perkins et al., 2007) is thus computed, which represents the sum of the lowest value of the two normalized histograms:

$$S_{index} = \sum min \left( normhist_{model}, normhist_{obs} \right) \tag{18}$$

where $normhist_{model}$ is the normalized histogram for models, the $normhist_{obs}$ is the normalized histogram for observations. A score is computed for each individual index and represents the degree of similarity between the model histogram and the observational histogram. The Perkins Skill Score of each index is then used to compute the DAV:

$$DAV = 100 * \frac{S_{GTI} - S_{index}}{S_{index}} \tag{19}$$

Where $S_{index}$ is the score obtained for the SPI, SPEI or Z-Score. A positive percentage denotes an added value from the proposed GDI index.

Finally, with the drought characteristics presented before, an assessment of the added value of the GDI against the SPI or SPEI, and Z-Score is performed, which allows to assess the biases difference for all land points and models:

$$av = (model_{index} - IB01_{index}) - (model_{DTI} - IB01_{DTI}) \tag{20}$$

Where the index denotes the SPI, SPEI or Z-Score indices. If the bias of the GDI is lower (higher), then the result would be positive (negative). This metric is computed for each land point and model and the final results are aggregated as boxplots. All metrics were computed for the 1989-2008 period.

## 3.    Results

### 3.1.    GDI general performance

All standardized indices (Figs. 2a to 2d) reveal percentiles close to those obtained by a normal distribution. Figure 2a displays the results for the GDI computed solely from precipitation. For the timescales below 90 days, the

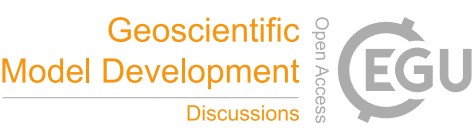

distributions deviate slightly from the normal, particularly for the 50th, 60th and 70th percentiles. While, for the longer aggregation scales, the time series deviates more from the normal distribution towards the tails. Figure 2b,
for the GDI computed with PR-PET, reveals a similar pattern. Yet, with the inclusion of PET, the deviations from the normal distribution are reduced. For SPI (Fig. 2c), the median values of each percentile align closely with the linear central line. The pattern of the spread from all distributions remains similar for all accumulation scales, with higher deviations towards the tails of the distribution. For SPEI (Fig. 2d), the inclusion of PET relative to the SPI (Fig. 2c) did not significantly improve the results as in Fig. 2b for the GDI and the pattern of spread
remains similar to SPI. Both SPI and SPEI present a larger spread in comparison to GDI. In the Z-Score index with PR (Fig. 2e) and PR-PET (Fig. 2d), the underlying distribution does not necessarily have to align with the normal standard distribution. Yet, due to the central limit theorem, for longer accumulations the time series tend towards normality, as evidenced by the alignment of the median values with the central line, despite the expected spread amongst all-time series.

As a complement, Fig. S3, in supplementary material, displays the statistics for all accumulation timescales and indices. The proximity of the values to zero indicates their similarity to the reference standard normal distribution. For the GDI indices, the mean and the median exhibit closer values, albeit with a slight deviation above 0. Nevertheless, those results together with the low skewness and kurtosis are good indicators of normality. For the SPI and SPEI indices, all metrics are near 0, although there is a more noticeable deviation among the time series
in comparison with the results obtained for the GDI. Conversely, by definition, the Z-Score displays a mean of 0 and a standard deviation of 1. As expected from the other statistics and the quantile-quantile plot from Fig. 2, the underlying distribution deviates from normality.

To evaluate the degree of similarity between the GDI against the other indices, the temporal correlation and the RMSE are presented in Fig. 3. Figure 3a shows the comparison between the GDI against the correspondent
SPI and SPEI, i.e., GDI(PR) is compared with SPI and GDI(PR-PET) compared to SPEI. Most of the cases reveal very high correlations, close to 1. The RMSE mirrors the results obtained for the correlations, with very low values, indicating a proximity of the index values across SPI or SPEI and the corresponding GDI. These outcomes suggest small differences amongst the indices for all timescales, i.e., GDI may detect the same drought events as SPI/SPEI. Fig. 3b displays the comparison between the GDI and the Z-Score, for both PR and the PR-PET as
input. Similar to Figure 3a, the correlations are notably high. Indeed, a high correlation is expected since the same time series were used for computing both the GDI and Z-Score. However, and for the short accumulations (7 and 15 days), the dispersion of the values is higher as given by the lower density of values (yellow and green, instead of red). For the RMSE, the prospect is different which is expected since the underlying distribution of the raw data is non-normal, hence larger RMSEs are found. The dispersion of the values is also greater in comparison to
the previous cases, as given by the lower densities. For the 7- and 15-day accumulations, the RMSE is approximately 0.4, gradually decreasing for longer accumulation periods. For the 180-, 360-, and 720-day accumulations, the underlying distribution of the Z-Score data tends to be closer to the standard normal, resulting in lower RMSE values and an approximation between GDI and Z-Score.



**Figure 2.** Quantile-Quantile plot against the 10th, 20th, 30th, 40th, 50th, 60th, 70th, 80th and 90th percentile from the standard normal distribution for all grid points of IB01 dataset for (a) GDI (PR), (b) GDI (PR-PET), (c) SPI, (d) SPEI, (e) Z-Score standardization (PR) and (f) Z-Score standardization (PR-PET). A smaller vertical spread indicates better agreement with the standard normal distribution. The horizontal line at each percentile denotes the median value across all locations.



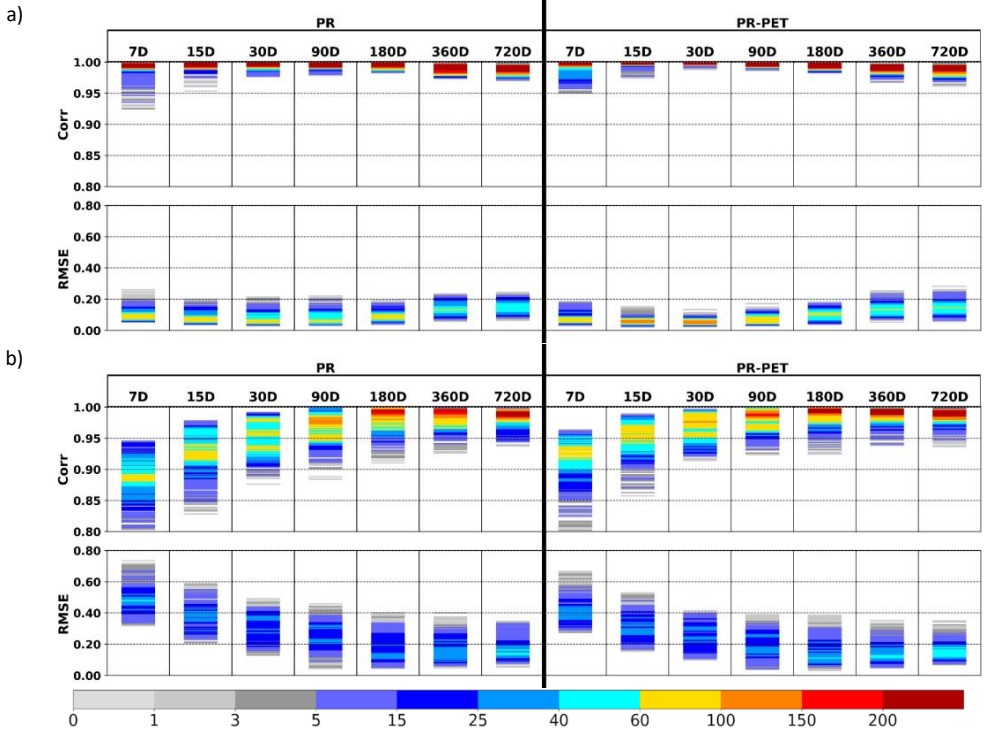

**Figure 3. Pearson correlation (top row) and Root Mean Squared Error (RMSE, bottom row) between (a) GDI index and SPI or SPEI and between (b) GDI index and Z-Score standardization Each column denotes the accumulation periods, where PR stands for accumulated precipitation (left) and PR-PET stands for accumulated precipitation minus potential evapotranspiration (right). The lines represent each land point for the IB01 observations. The different colors are a measure of density given by counting the number of occurrences within each horizontal strip with a thickness of 0.001 for both the correlation and RMSE.**

When evaluating the performance of a drought index, it is crucial to assess key characteristics such as event severity, frequency, and duration. Table 2 displays the spatial correlation and RMSE for drought intensity, frequency, and duration between the GDI against the SPI or SPEI (Table 2a) and against the Z-Score index (Table 2b). Only the results for moderate drought are considered, since for higher drought thresholds, the lack of events may hinder this comparison. Overall, the spatial comparability of the intensity and mean duration of droughts characteristics, between the GDI and SPI or SPEI (Table 2a) decreases toward higher accumulations, as indicated by the declining correlations and larger RMSEs. Those results are opposed to the findings from Figure 3, where correlations remained high and RMSE low for all timescales. As for the frequencies, although the correlation decreases for higher aggregation periods, the RMSE also decreases. Nonetheless, for all cases, the higher the base value for each drought characteristic and accumulation, shown Fig. S4, S5 and S6, imply larger differences amongst the indices. For the comparison with the Z-Score index (Table 1b), the results are similar to Table 1a. Still, some differences arise, namely at the 7-day accumulation for drought intensity and frequency, where the correlation and RMSE do not follow the pattern described earlier.



**Table 2. Spatial Pearson Correlation (top left for each cell) and spatial Root Mean Squared Error (bottom right for each cell) between (a) GDI against SPI or SPEI and (b) GDI against the Z-Score index. Each row, for both panels and from top to bottom, represents the drought intensity, drought mean decadal frequency and drought mean duration. These results are presented only for moderate drought (index< -0.5). Each column denotes the accumulation periods, where PR stands for accumulated precipitation (left) and PR-PET stands for accumulated precipitation minus potential evapotranspiration (right).**

a)

| | | PR | | | | | | PR-PET | | | | | |
|---|---|---|---|---|---|---|---|---|---|---|---|---|---|
| | 7D | 15D | 30D | 90D | 180D | 360D | 720D | 7D | 15D | 30D | 90D | 180D | 360D | 720D |
| inte | 0.95 / 0.53 | 0.96 / 0.75 | 0.92 / 1.58 | 0.79 / 3.74 | 0.8 / 6.02 | 0.75 / 16.45 | 0.75 / 38.63 | 0.98 / 0.37 | 0.97 / 0.52 | 0.95 / 1.12 | 0.9 / 3.06 | 0.83 / 8.65 | 0.77 / 20.53 | 0.72 / 45.68 |
| Eve | 0.94 / 5.0 | 0.96 / 3.52 | 0.93 / 3.77 | 0.81 / 3.69 | 0.82 / 2.36 | 0.82 / 2.27 | 0.86 / 2.13 | 0.97 / 4.19 | 0.97 / 2.95 | 0.94 / 2.77 | 0.88 / 2.46 | 0.84 / 2.45 | 0.82 / 2.45 | 0.86 / 2.29 |
| Days | 0.93 / 0.42 | 0.95 / 0.64 | 0.9 / 1.27 | 0.77 / 2.82 | 0.77 / 5.52 | 0.73 / 16.29 | 0.73 / 34.53 | 0.93 / 0.32 | 0.96 / 0.48 | 0.93 / 1.0 | 0.88 / 3.01 | 0.81 / 8.13 | 0.74 / 19.3 | 0.69 / 40.86 |

b)

| | | PR | | | | | | PR-PET | | | | | |
|---|---|---|---|---|---|---|---|---|---|---|---|---|---|---|
| | 7D | 15D | 30D | 90D | 180D | 360D | 720D | 7D | 15D | 30D | 90D | 180D | 360D | 720D |
| inte | 0.71 / 3.64 | 0.85 / 3.81 | 0.85 / 4.3 | 0.72 / 5.05 | 0.76 / 7.04 | 0.71 / 16.13 | 0.76 / 35.96 | 0.8 / 2.8 | 0.85 / 2.96 | 0.87 / 3.43 | 0.85 / 4.87 | 0.81 / 8.9 | 0.78 / 20.73 | 0.74 / 44.58 |
| Eve | 0.92 / 8.15 | 0.96 / 4.39 | 0.93 / 3.82 | 0.78 / 3.8 | 0.8 / 2.45 | 0.79 / 2.38 | 0.86 / 2.16 | 0.94 / 8.62 | 0.96 / 4.19 | 0.94 / 2.92 | 0.85 / 2.78 | 0.82 / 2.62 | 0.81 / 2.57 | 0.85 / 2.4 |
| Days | 0.9 / 0.53 | 0.92 / 0.76 | 0.88 / 1.33 | 0.71 / 3.23 | 0.75 / 5.92 | 0.68 / 17.88 | 0.74 / 33.85 | 0.94 / 0.5 | 0.93 / 0.77 | 0.91 / 1.28 | 0.84 / 3.62 | 0.79 / 8.49 | 0.76 / 19.62 | 0.71 / 39.22 |

For instance, in case of the correlation for drought intensity, a value of 0.71 is attained for the PR indices and 0.8 for the PR-PET indices, contrasting from the 0.95 and 0.98 obtained in the same comparison of the GDI against the SPI or SPEI, respectively.

### 3.2.    2004/2005 case study

A case study of an extreme drought that affected the Iberian Peninsula, starting in the autumn of 2004 is presented in Fig. 4, where the spatial extent for moderate, severe, and extreme drought are shown, computed for all timescales and indices. For this analysis, only the SPEI, and the Z-Score together with the GDI based on PR-PET for the IB01 dataset is considered. Figure S7 in the supplementary material shows the same but for the PR-based indices. In terms of drought spatial extent, the difference between PR and PR-PET based indices is minimal. This event is widely regarded as one of the most extreme droughts in recent history affecting the Iberian Peninsula, characterized by a precipitation deficit during the hydrological year of 2004/2005 (García-Herrera et al., 2007; Santos et al., 2007). The patterns of the spatial extent for all drought severities for the lower accumulations (Figs. 4a, 4b and 4c) are similar, revealing the extreme dry autumn of 2004 and spring of 2005, which somewhat repeated in the following year. At these scales, the standardized indices exhibit similar percentages of territory in the three drought categories.

The SPEI and GDI show that approximately 75% of the land experienced extreme drought conditions in October 2004 to May 2005. On the contrary, the Z-Score stands out with lower spatial percentages of Iberia experiencing severe drought and almost no territory is classified as extreme drought. As for the longer accumulations (Figs. 4d to 4g), all three indices converge in terms of the spatial extent of drought. At 90- and 180-days accumulation almost the entire year of 2005 is at least in moderate drought, returning to normal conditions in 2006. For the 360-day accumulation, the peak of drought severity occurs during the summer and autumn of 2005. It is worth noting that summer is typically one of the most critical periods for drought due to reduced precipitation and increased temperatures. For the 720 days, however, the drought conditions started in 2005 with a peak in 2006, revealing a shift relative to the previous cases, due to the longer accumulation. As indicated by the results from Figs. 4a to





4c, the autumn of 2005 and spring of 2006 were also drier, albeit not as severe as during the previous hydrological year.

Figure 5 shows the time series of the spatial means of the PR-PET indices for the same period and event as Fig. 4 (October 2004 to September 2006). Figure S8 shows the same, but for the PR based indices. The red and blue shadings denote the differences between the GDI (solid black line) and the SPEI and the Z-Score, respectively. Amongst all aggregation periods (Figs. 5a to 5g), the differences across indices are more visible towards the extreme values. Overall, the differences are larger for the Z-Score rather than for the SPEI. Since the Z-Score is based on a simple standardization, it closely follows the accumulated PR-PET patterns, while the same could not be applied to the standardized indices, hence the higher differences. Still, for the lower aggregation periods (Figs. 5a to 5c), the Z-Score fluctuates more relative to the SPEI and GDI. As for the GDI, the day-to-day values can reach higher extremes. Since the index is based on an empirical distribution, the maximum and minimum of the time series is dependent on the length of the data used. On the other hand, for indices such as the SPEI, the extremes are more controlled by the chosen distribution, while for the Z-Score, the extremes are dependent on the difference of the accumulations relative to their mean. Those factors cause a smoothing effect on the extremes for the SPEI, SPI and Z-Score.

### 3.3. Performance with regional climate model data

To assess the performance of the new index, the DAV metric (Eq. 19) is applied (Table 3). The Perkins Skill Scores are determined for GDI, SPI, SPEI and Z-score with the EURO-CORDEX models and IB01, and the results are aggregated for all-time series. Table 3a shows the DAV for GDI with PR against SPI, and GDI with PR-PET against SPEI. For each accumulation period, a histogram is built representing the percentage of land points within each DAV category. The vast majority of locations for all accumulations clearly displays positive added value. For the shorter accumulation periods (less or equal to 30 days), the results indicate that approximately 25 % of PR indices and 50 to 60 % of points of the PR-PET indices exhibit roughly a neutral DAV (-5 to 5 %). Moreover, negative DAV, below -5 % only occurs for short accumulation in isolated cases. Most of points outside the neutral DAV range reveal important added value, with DAV between 5 and 10 % representing the majority of the cases, with more than 40 % of grid points, in particular for the intermediate accumulations of 30-, 90- and 180-days.

For accumulations of 180-, 360-, and 720-days, there is a greater dispersion of the DAV percentages towards larger added value with cases reaching a DAV of 60 %. For those long accumulation scales, less than 3 % of all land points reveal neutral DAV for the 180 days, further decreasing for the longer accumulation scales. GDI can add value, most likely due to its improved representation of the shape of the cumulative distribution and consequently of the normality, as indicated in Fig. 1 and S2. Table 3b displays the DAV metric between the GDI and the Z-Score, showing that the GDI index reveals again a positive added value. Despite the similarities with Table 3a, larger DAV values occur for a higher percentage of land points, however the differences are still small.





**Figure 4. Time series of the daily drought spatial extent of the period from 1 October 2004 until 30 September 2006 for the IB01 with aggregations of (a) 7-, (b) 15-, (c) 30-, (d) 90-, (e) 180-, (f) 360- and (g) 720-days. In each panel the top row displays the results for the GDI index, the middle row for the SPEI index and the bottom row for the Z-Score standardization. All indices consider only the balance between precipitation and PET. The yellow color denotes the results for moderate drought for index <-0.5, light orange for severe drought for index <-1 and dark orange for extreme drought for index <-1.5.**



**Figure 5.** Area average time series of the daily drought indices for the period from 1 October 2004 until 30 September 2006 for the IB01 dataset with aggregations of (a) 7-, (b) 15-, (c) 30-, (d) 90-, (e) 180-, (f) 360- and (g) 720-days. The black line represents the GDI index, while the red and blue shadings denote, respectively, differences between the GDI and SPEI and between GDI and the Z-Score index. The differences are computed by taking the absolute value of each index first. Thus, if the difference is positive, it implies that the GDI is further away from the mean 0 than the other index. All indices only consider the balance between precipitation and PET.



**Table 3. Distribution Added Value for (a) GDI against SPI or SPEI, (b) GDI against the Z-Score index. Each column denotes the accumulation periods, where PR stands for accumulated precipitation (left) and PR-PET stands for accumulated precipitation minus potential evapotranspiration (right). The colors and values in each cell correspond to the percentage of land points within the respective category. The Perkins Skill Score is built by confronting each model against the observations in terms of their distribution. The DAV is then computed as the relative difference between each index Perkins Skill Score. The DAV is computed for each model and each location individually. Therefore, the table shows the aggregation of the results obtained for all models.**

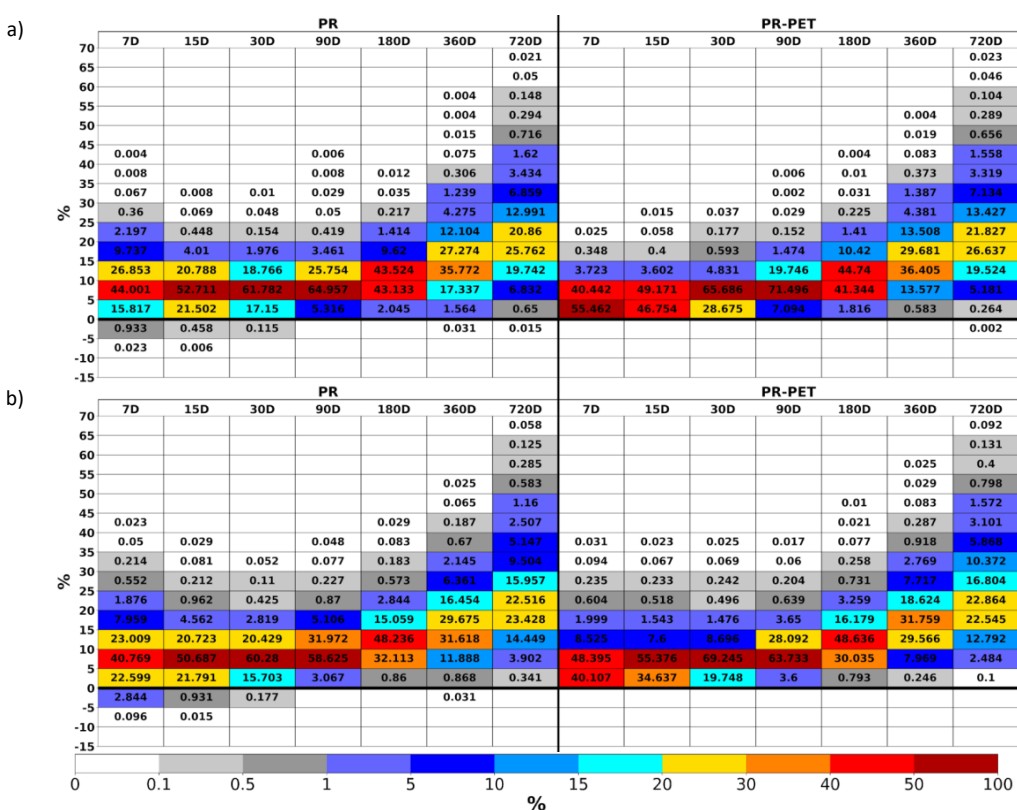

These findings are relevant and could indicate that the fit of the data to a theoretical distribution could introduce some uncertainty into the final index, while a fit to an empirical distribution can approximate the modelled and observational distributions.

Regarding the gains for drought characteristics, Fig. 6a presents the bias difference for the intensity, frequency, duration, and spatial extent between the GDI against the respective SPI or SPEI (Eq. 20). If the bias obtained comparatively to the SPI or SPEI is lower, then the resulting value will be positive, meaning an add value. Each boxplot features the results from all models and locations, subdivided as moderate (yellow), severe (orange) and extreme (red) drought conditions. It is important to note that the extreme drought class is contained within severe drought, which by its turn are contained within moderate drought. This approach avoids splitting the events (Soares et al., 2023b). The position of the median in this context is relevant, as positive values indicate an added value from more than half of all time series. For the mean event severity not only does the dispersion of the results increase towards the long accumulations, but the median value is also higher, tending to be more positive. Those





findings indicate higher gains for more sites. Moreover, the differences in the performance across the drought
severities are small, but the spread tends to be larger for extreme drought for accumulation below 180 days. It is
worth noting that the scale is not linear, thus for higher values the differences in the spread may not be as
noticeable. Additionally, the inclusion of the PET reduces the difference in performance. As for the decadal
frequency, the median is mostly neutral, except for short accumulations of 7 and 15 days, while the spread tends
to decrease for higher accumulations. In this case, the difference between PR-based or PR-PET balance-based
indices is negligible. Similarly, the dispersion among the three drought types is small. For the mean event duration,
a similar pattern is shown in comparison to the intensity. The median values are mostly positive for accumulations
above 180 days. For lower timescales, the bias differences are more centered around 0. Once again, the dispersion
of the values increases for higher accumulations. Regarding the spatial extent of drought, the bias difference
pattern is similar for all timescales, with the main distinction focused on the different drought types. In this case,
the spatial extent is similar amongst the indices.

Figure 6b illustrates the same bias difference as Fig. 6a, but in this case, comparing the GDI against the Z-Score
standardization. Regarding the evaluation of the drought characteristics, it is important to consider that those are
set to 0 in case the index never falls below the defined drought threshold. If this occurs for both models and
observations, then the corresponding location is set to a missing value, allowing a correct assessment of the
boxplots since the missing values are not included. In contrast, for the standardized indices, this issue usually does
not arise. For all drought characteristics, the range of the boxplot is similar to Fig. 4a, reaffirming the negligible
difference between the PR based and the PR-PET based indices. However, some differences relative to Fig. 6a
still occur. Firstly, the spread of the time series bias difference tends to be larger in Fig. 6b. On the other hand,
for most cases, the median values are also higher compared to Fig. 6a. Moreover, highlight to the mean event
severity and duration for the 7-days accumulation featuring a clear gain in values for extreme drought. The
exceptions are the negative values for the severe and extreme drought spatial extent. Yet those negative values
for the 7 days progressively increase as the accumulation period increases. This increase can be attributed to the
Z-Score's approximation to normality for higher accumulations (Figs. 2 and S3). For the lower timescales, the Z-
Score underrepresents the extreme drought for the selected thresholds, leading to fewer events in comparison with
the other indices, as hinted by Fig. 4. With lower percentages, the model to observations bias is also reduced,
hence the negative values. Figure S9 shows the absolute model to observations bias boxplots for the SPI and SPEI,
Z-Score and GDI indices. The three figures are similar, with the exception of the drought spatial extent for the Z-
Score. In fact, this underrepresentation of extreme drought also affected the results in Fig. 6b. As evidenced in
Fig. 4, the Z-Score of the IB01 reveals a lower spatial extent in drought, thus the difference between simulations
and observations are smaller in comparison to other statistics (Fig. 4).

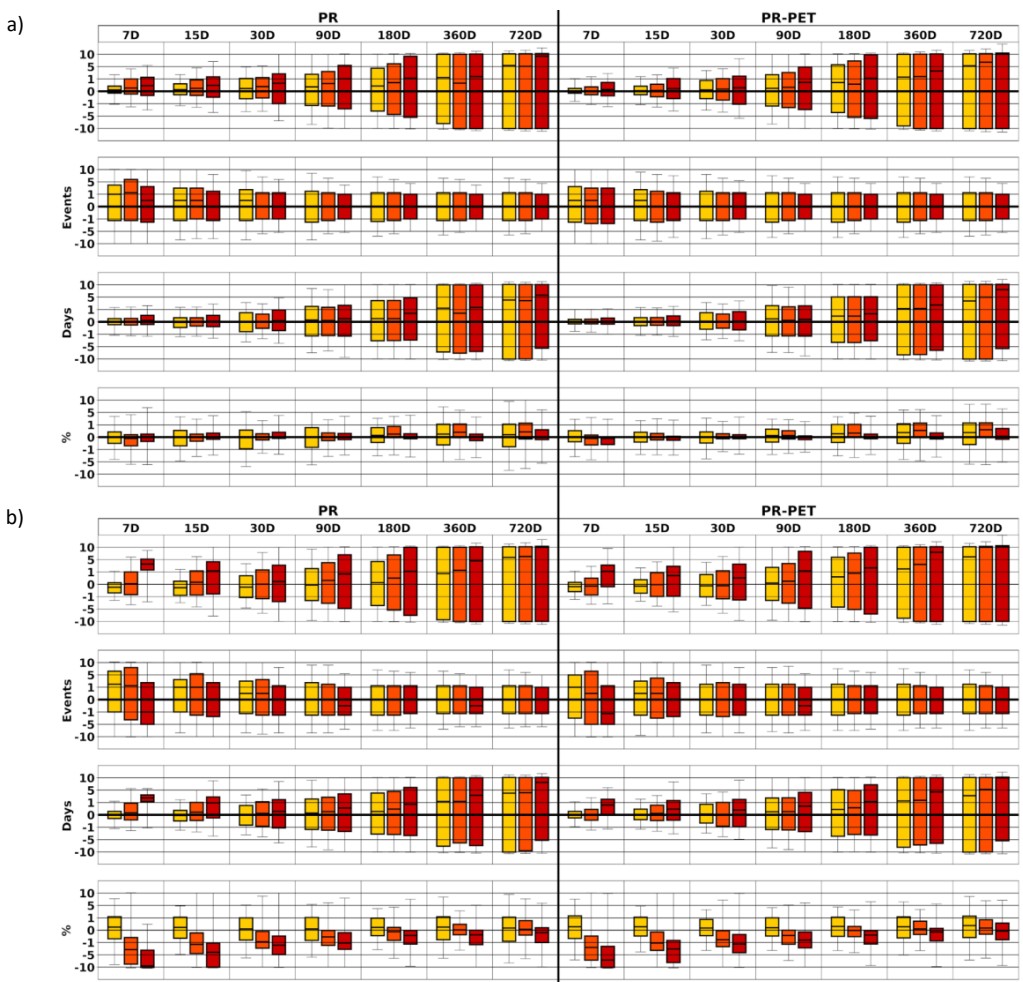

**Figure 6. Boxplot of all land points featuring the bias difference between the (a) GDI index and SPI or SPEI and between (b) GDI index and Z-Score index. The bias difference or av is computed following Eq. 20. Both EURO-CORDEX models and IB01 observations were considered. In each sub-figure, from top to bottom: mean event severity, decadal frequency, mean event duration and spatial extent. Each column denotes the accumulation periods, where PR stands for accumulated precipitation (left) and PR-PET stands for accumulated precipitation minus potential evapotranspiration (right). The different boxplot denotes the results for moderate drought for index <-0.5 (yellow), severe drought for index <-1 (light orange) and extreme drought for index <-1.5 (dark orange). For each boxplot the low (high) whisker denotes the 1st (99th) percentile, while the three horizontal lines within the box correspond, from bottom to top to the 25th, 50th and 75th percentiles.**

## 4.  Discussion and Conclusions

In the present study, a new drought index is introduced, designated as Generalized Drought Index or GDI. This index is extremely straightforward to compute since the fitting process to a distribution is not required. It is empirically based and can be regarded as an alternative to other indices such as the SPI or SPEI. The GDI is computed by generating an empirical distribution based on a smoothed cumulative histogram, where the PR or



PR-PET data can be converted into probabilities and then brought back as standardized values following a normal distribution of mean 0 and a standard deviation of 1.

As an example, the IB01 observational-based dataset and the evaluation EURO-CORDEX simulations were used to compute the GDI for Iberia. The analysis period was limited to the EURO-CORDEX available period, from 1989 to 2008. The use of both models and observations serves two purposes: to facilitate the comparison and assessment of GDI against a daily version of the SPI and SPEI indices, along with a simple Z-Score standardization; and the inclusion of modeled data also allows an assessment of the added value with the DAV

metric. Still, only the IB01 dataset is considered for the main comparison across all indices.

The main distinction between GDI and SPI or SPEI lies in the fitting of climate data to a probability distribution function. In the case of the monthly SPI and SPEI, the fitting process is mandatory due to the scarcity of data points compared to daily data, which in turn hinders the applicability of the GDI. Nevertheless, in this study, and since the seasonal cycle was previously removed, both the daily SPI and SPEI were fitted to the log-logistic

distributions. A reason to consider the log-logistic distribution for SPI lies in the poor fit found for the Gamma distribution. It is important to note that assuming a specific distribution for different climate types or even different accumulations can be detrimental since the underlying distributions may differ (e.g., Stagge et al., 2015; Monish and Rehana, 2020; Zhang and Li 2020). Those issues do not arise with the GDI index. Furthermore, SPI and SPEI may encounter difficulties in fitting data from semi-arid and arid climates with very low accumulations, such as

deserts (Beguería et al., 2014). The very low accumulations, which could also occur in climate change studies with historical reference, may not be very well simulated, although alternative approaches exist such as Spinoni et al. (2018) and Soares et al. (2023b). On the other hand, the GDI does not have those constraints, since the distribution is built empirically, through smoothing an accumulated histogram of the data. Apart from the distribution, the definition of the parameters in the case of SPI and SPEI may also pose challenges, especially at

the daily scale. Outliers present in the data may hinder the parameter estimation process. Additionally, as the dataset size increases, the computational expense for calculating the index also escalates, particularly when considering methods such as maximum likelihood (Beguería et al., 2014). In contrast, the GDI offers a simple and computationally efficient procedure which easily contours the determination of fitting parameters, making it suitable for large datasets. The performance of the GDI may be superior for larger datasets, as the underlying

distribution associated with each location is better defined. Indeed, the GDI time series conforms better to the theoretical standard normal distribution, namely if the PET is considered. Another potential source of uncertainty, particularly for extreme values, is associated with the Abramowitz and Stegun (1965) approximation, which performs better for values closer to the mean. For extremes, the values deviate from the true standard normal distribution, although the likelihood of SPI or SPEI being extreme is low (Stagge et al., 2015). GDI does not

exhibit this limitation, due to its empirical nature, the minimum and maximum value of the index is dependent of the size of the time series and its values do not deviate from the standard normal as much as the other indices.

In order to assess the degree of similarity amongst the indices, a Pearson correlation and a RMSE were computed. The results reveal a strong agreement among the three indices, particularly between the GDI with SPI or SPEI. Nevertheless, lower correlations and higher RMSE occur for the comparison to the Z-Score. In this case, those



deviations are expected since the original underlying distribution is kept for the Z-Score index. Regarding the drought characteristics such as intensity, frequency, and duration, for moderate drought conditions, the GDI shows similar results against the SPI, SPEI and Z-Score. Still, the proposed index reveals differences, since for most cases and towards longer aggregation periods, the similarity amongst indices decreases.

The GDI is also evaluated in terms of drought spatial extent against the SPEI and Z-Score for the severe 2004/2005
drought event. While the standardized indices reveal very similar values for all timescales and drought severities, the Z-Score had some difficulty in representing severe and namely the extreme drought for the lower accumulations. On the contrary, all the three indices converge due to the approximation towards normality for longer accumulations. Regarding the performance of the GDI, the proposed index demonstrated a close representation of the spatial extent of drought in comparison with the SPEI. The same conclusions can be drawn
for the PR-based indices. As for the spatial mean time series the indices tend to grow apart towards the extreme values. Nevertheless, the differences are small, in particular for the comparison of GDI against the SPI/SPEI. Those findings provide reassurance and present the GDI as a viable alternative to the SPI or SPEI indices.

Regarding the evaluation of the proposed metric with the DAV, the GDI exhibits a positive added value against the other drought indices. The gains of the proposed index against the SPI and SPEI are relevant since both indices
already conform to the standard normal distribution. Those findings could potentially indicate a reduction of the uncertainty due to the fitting procedures. Thus, this added value is not solely owned to the standardization, otherwise the DAV would be closer to 0 % in this case. As for the comparison against the Z-Score, the GDI displays higher DAV percentages, highlighting the importance of both models and observations of having the same underlying distributions, when comparing different drought indices.

The Z-Score synchronizes only the statistical mean and standard deviation, by setting them to 0 and 1 respectively, disregarding the underlying distribution. In contrast, GDI, SPI or SPEI share the same distribution and parameters: a normal distribution with a mean of 0 and a standard deviation of 1. PR or even PR-PET tend to be positively skewed which may result in an underrepresentation of severe and extreme drought for shorter accumulations on threshold-based definitions. For the standardized indices, since they follow the standard normal distribution, this
issue does not arise. Nevertheless, the GDI adds value to at least half or more locations for all drought characteristics. This occurs for both the comparison against the SPI and SPEI and against the Z.-Score. The only exception where a clear loss of value is observed occurs for severe and extreme drought spatial extent for the lower accumulations.

The GDI can identify the same events as the SPI or SPEI by returning similar results for most cases, while
revealing an enhanced performance over the Z-Score index. Yet, it is less computationally expensive than SPI and SPEI, does not need the assumption of the characteristics of the underlying distribution of the data and can be appliable to other variables such as soil moisture or actual evapotranspiration. Comparatively to other studies also featuring a daily index (e.g., Li et al., 2020; Ma et al., 2020, Zhang et al., 2022a), a daily approach to GDI offers a finer temporal resolution, enabling a more detailed depiction of meteorological variations and their
immediate impacts on drought conditions. The higher sensitivity in comparison to monthly indices allows for the timely identification of short-term drought events, better capturing the start and duration. This feature is

particularly relevant in regions characterized by a large climatic variability. Thus, GDI is suitable to characterize meteorological, hydrological, and agricultural droughts with the same methodology. Therefore, GDI could be a viable alternative for computing a standardized index for drought for large kilometer-scale simulations such as

those from the WRCP Flagship Pilot Study on "Convective phenomena over Europe and the Mediterranean" (Coppola et al., 2019).

Nevertheless, some questions are still open. Firstly, how different are the GDI results for climate change projections? The proposed index is empirically based, thus the approach from Spinoni et al. (2018) or Soares et al. (2023b) can be considered for climate change assessment, where the entire time series is considered to build

the empirical distribution. However, would still be possible to use the historical period as reference, conserving the absolute values for both the reference and future periods? What is the applicability and performance of the index at a global level? Other questions also arise, namely the sensitivity of the index to the potential evapotranspiration method considered and the sensitivity related to the initial variables. Furthermore, can the index be used for an ensemble-based analysis? Some of these questions will be pursued in future studies.

**Data availability.**

All model and observational datasets are publicly available. The regional and global model data are available through the Earth System Grid Federation portal (Williams et al., 2011; https://esgf.llnl.gov/, last access: September 2023). The Iberia01 dataset is publicly available through the DIGITAL.CSIC open science service (Herrera et al., 2019a, https://doi.org/10.20350/digitalCSIC/8641).

**Author contributions**

João Careto developed the new drought metric the GDI, designed and wrote the paper. Rita Cardoso, Ana Russo, Daniela Lima and Pedro Soares provided useful insights and made major contributions to the writing process. All authors read and approved the final manuscript.

**Competing interests.**

The contact author has declared that neither they nor their co-authors have any competing interests.

**Acknowledgements**

The authors would like to thank all the individual participating institutes listed in Tables S1 and the Earth System Grid Federation infrastructure for providing all the model data used in this study. The authors also acknowledge the Iberian Gridded Dataset (IB01) http://hdl.handle.net/10261/183071 (Last access: 1st August 2023).



**Funding**

This work was supported by the Portuguese Fundação para a Ciência e a Tecnologia (FCT) I.P./MCTES through national funds (PIDDAC) – UIDB/50019/2020- IDL, DHEFEUS (https://doi.org/10.54499/2022.09185.PTDC). JC, RMC, AR and DCAL are supported by the Portuguese Foundation for Science and Technology (FCT) financed by national funds from the MCTES through grants SFRH/BD/139227/2018,
https://doi.org/10.54499/2022.01167.CEECIND/CP1722/CT0006 and https://doi.org/10.54499/2022.03183.CEECIND/CP1715/CT0004, respectively. Pedro MM Soares would like to acknowledge the financial support of FCT through project UIDB/50019/2020 (IDL and EEA-Financial Mechanism 2014–2021) and the Portuguese Environment Agency through Pre-defined Project-2 National Roadmap for Adaptation XXI (PDP-2).




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
