# Peer review of "Generalised drought index: A novel multi-scale daily approach for drought assessment"

_Geoscientific Model Development, 2024_

## Author Response (AR1)

**First Reviewer**

**General Comments:**

A new daily based drought index is proposed in this study, estimated over the Iberian Peninsula and evaluated his added value with respecto to otgher existing drought index. To this aim, an observational gridded datataset and regional models have been used. The paper is well written and the results are well explained, including figures and tables. Considering the region of analysis, and its expected challenges in a climate change context, the work is very relevant, timely and important to improve the drought monitorization over the Iberian Peninsula. The main critical point I have identified could be the use of another observational dataset as reference to better define the frmaework. Based on these points wy recommendation is accepted with minor changes or, if the new observational reference is added, major changes.

We wish to thank the reviewer for the insightful and constructive feedback on our study. We sincerely appreciate your thorough evaluation of our proposed daily-based drought index (GDI, Generalized drought index), particularly the consideration of its relevance and timeliness. We are grateful for your positive remarks, and we believe that accordingly we were able to improve the manuscript. We have noted your suggestion regarding the use of another observational dataset as a reference to enhance the framework of our study. However, using different base datasets would not allow a fair comparison of the GDI against other drought indices. Therefore, instead, and understanding your comment, we compared the traditional monthly SPI and SPEI indices computed with the default settings with the R package "SPEI" and R environment. These results now will be available on the supplementary material.

**Comments:**

**-** Why has the added value evaluated with the Evaluation experiment of CORDEX? I understand that this analysis is relevant in order to apply this index in a context of climate change but I am not sure that the added value justifies the increment of the paper and analysis.

The initial idea was to understand if the new index would change the relationships between models and observations, in comparison with the SPI, SPEI and even Z-Score. At the same time and in an indirect form, the added value metric could indicate that the underlying distribution of the GDI was better able to conform to the standard normal distribution. However, and acknowledging your important comment, we now realise that the use of EURO-CORDEX in this context could be misleading to the reader. Therefore, we decided to remove this model dataset and focus only on the proposed index with the Iberian gridded dataset, simplifying the paper's overall methodology. This implies some changes, in particular for those figures in which the EURO-CORDEX was considered, namely Table 3 and Figure 6. For Table 3 (DAV results) the ideal or reference distribution is now the standard normal distribution itself. Thus, if the GDI can obtain a higher score compared to the other metrics, it will result in a positive value, meaning that the GDI was able to improve the results. Figure 6, changes from an added value perspective to the differences expected across the various methodologies for the drought characteristics.

- Are 20 years enough to make a proper normalization?

Indeed 20 years of daily data might not be sufficient for a proper normalization. However, with the removal of the Euro-CORDEX dataset, we now used the full period available for the Iberian Gridded dataset (1971-2015, i.e. 45 years). Therefore, all figures and tables were remade to account for the new period. Moreover, some reordering of the sections was performed, namely former section 3.2 (case study now is the last section 3.3, and former section 3.3 now is 3.2 (and changed its name to: "Drought Characteristics Assessment"). Further changes include new text for the now table 2 and 3, figures 3 and 4. On Figure 2 the same conclusion still hold, and for figures on the 2005 case-study no changes were made to the text. Throughout the manuscript other small changes and corrections were made and in some cases to also accommodate the suggestions of reviewer 2.

- Lines 47-50: However is a proper link between these two sentences?

"… Nevertheless, the consideration of just precipitation by SPI could be a limiting factor depending on the climatic dominating conditions in certain regions. However, with anthropogenic climate change, rising temperatures and the subsequent increases in evapotranspiration can also significantly increase the impact of drought events (Hu and Wilson, 2000; Vicente-Serrano et al., 2010)…."

Corrected. We changed "however" to "Moreover", We thank the reviewer for noting this mistake.

- Line 77: Something is missing here? ".. and is also considered in the by the Reconnaissance Drought Index (Tsakiris and Vangelis, 2004)."

We thank the reviewer for noting the mistake, which we corrected to "and is also considered in the standardised Reconnaissance Drought Index".

- There is a global SPI and SPEI gridded observational dataset publicly distributed (Vicente-Serrano et al 2010a and 2010b), that could be used as reference and/or to analyze the obvservational uncertainty of the results. However, the authors decided to use the evaluation experiment of CORDEX, is there any reason? I didn't see it (probably I missed the section in which the authors explain it) in the manuscript and I am not sure what is the advantage when a new index is evaluated.

Vicente-Serrano S.M., Beguería S., López-Moreno J.I., 2010 (a): A Multi-scalar drought index sensitive to global warming: The Standardized Precipitation Evapotranspiration Index – SPEI. Journal of Climate 23(7), 1696-1718, DOI: 10.1175/2009JCLI2909.1.-- http://digital.csic.es/handle/10261/22405.

Vicente-Serrano S.M., Beguería S., López-Moreno J.I., Angulo M., El Kenawy A. (2010(b)) A global 0.5° gridded dataset (1901-2006) of a multiscalar drought index considering the joint effects of precipitation and temperature. Journal of Hydrometeorology 11(4), 1033-1043, DOI: 10.1175/2010JHM1224.1.-- http://digital.csic.es/handle/10261/23906

We thank the reviewer for this nice suggestion, and we are aware that such a database exists. However, the SPEIBASE is based on the CRU dataset, which is different from the Iberian Gridded Dataset and has a rather low resolution (0.5°) for the Iberian climate heterogeneities. Thus, in our opinion, any comparison would be difficult to perform since we were dealing with different indices and different baselines series to produce the indices. Nevertheless, we agree with the reviewer that a comparison with a 'default' and widely used SPI or SPEI at the monthly time step would be meaningful. Therefore, we added a second analysis throughout the supplemental material featuring the daily or monthly GDI (depending on the

situation) and the monthly SPI and SPEI computed with the IB01 and using the 'SPEI' package from R (Beguería et al., 2014) in its default configuration. For the monthly GDI, whenever it is applied, the daily index is first aggregated at the monthly time scale by simple averaging. The results reveal that aggregating the daily index GDI at the monthly time-step, would degrade the comparability against the monthly SPI and SPEI, namely at the shorter accumulations. Therefore, larger differences occurred at the 1-month timescale. Nevertheless, the comparison with the DAV (Table S1) reveals that even when the GDI is aggregated at the monthly time-step, there is still added value. In this context, the concept of added value is used to quantify the similarity of distributions to the standard normal.

Beguería, S., Vicente-Serrano, S. M., Reig, F., and Latorre, B.: Standardised precipitation evapotranspiration index (SPEI) revisited: parameter fitting, evapotranspiration models, tools, datasets and drought monitoring, International journal of climatology, 34(10), 3001-3023, doi:10.1002/joc.3887, 2014.

- Line 187: Which is the final period? 1989-2009 or 1989-2008

We removed this sub-section from the manuscript. Now, the final period is the full range from the IB01 dataset: 1971-2015.

- Lines 185-189: EuroCORDEX also contains future projections with different scenarios.

We thank the reviewer for the comment. We are well aware of this fact. Nevertheless, and due to the removal of the EURO-CORDEX, this entire sub-section was removed from the manuscript.

- Line 190: The evaluation experiment is partially synchronized as it inherit the temporal correspondence of the reanalysis.

This sub-section was removed.

- Lines 224-225: The Iberian Peninsula presents some regions with arid and semiarid climates so this sentence could be modified accordingly: instead "*Therefore, in the context of climate change, this equation is not the best option for computing PET (Beguería et al., 2014).*" the authors could write something like "*Therefore, in the context of climate change and the Iberian Peninsula, with arid and semiarid regions, this equation is not the best option for computing PET (Beguería et al., 2014).*" which is more explicit and reflects the particular problem of the Iberian Peninsula.

We thank the reviewer for this nice suggestion, and we changed it accordingly.

- Subsection 2.4. GDI evaluation: If the GDI is obtained with an histogram estimation, could the evaluation based on histograms bias the results in favour to this index?

Indeed, the index is computed based on a histogram. However, this initial histogram is built to closely match the original time series. Subsequently, we apply a cubic spline to smooth the histogram. Afterwards, the data is fitted to this spline distribution in order to obtain the index. On the other hand, the evaluation of the index also assumes the building of a histogram, with bin limits determined by the reference distribution (standard normal distribution), which differs from the previous case. Thus, we believe that there is no favouring towards the proposed index, as not only the building of the histograms

is fundamentally different, but also considering the results obtained by the q-q plot (Figure 2 of the main manuscript) clearly indicate an improvement achieved by the GDI.

- How are the possitive percentages of DAV translated to added value with respect to the corresponding index?

In the submitted version, positive DAV means that the distribution of the observational reference and the distribution of the model (for a specific location) are closer to each other for the GDI, than for any of the other indices. However, to simplify the manuscript and avoid confusion, we removed the EURO-CORDEX data as mentioned before. Still, we wanted some sort of comparison or quantification of the added value of the new index against the established ones. Thus, we now considered the standard normal distribution as reference (Table 2 of the revised manuscript), and if the GDI reveals a positive DAV, then that means it better conforms to the standard normal distribution than SPI or SPEI.

- Table 2 is a figure? Why had the autors not used an standar table?

Table 2, now Table 3 in the revised manuscript can be considered a table, It appears as a figure because it was made with the programming language Python.

**Second Reviewer**

We wish to thank the reviewer for the insightful and constructive feedback on our study. We sincerely appreciate your thorough evaluation of our proposed daily-based drought index (GDI, Generalized drought index). We are grateful for your positive remarks, and we believe that we were able to improve the manuscript.

1. Abstract: The region of study (Iberian Peninsula) is not very clearly stated. In fact, as EuroCORDEX is mentioned, it could be understood that the whole Europe is the area of analysis

We thank the reviewer for the comment. Indeed, the use of the EURO-CORDEX dataset might be misleading to the reader. Based on a comment made by the first reviewer, and this one of yours, we decided to remove this dataset and focus only on the proposed index with the Iberian gridded dataset only, allowing a simplification of the overall methodology of the paper. This implies some changes, in particular for those figures in which the EURO-CORDEX was considered, namely Table 3 and Figure 6. At the same time, with the removal of the EURO-CORDEX dataset, we now used the full period available for the Iberian Gridded dataset (1971-2015), 45 years instead of 20 years. Therefore, all results, figures and tables were remade to account for the new period. Moreover, some reordering of the sections was performed. Further changes include new text for the now table 2 and 3, figures 3 and 4. On Figure 2 the same conclusion still hold, and for figures on the 2005 case-study no changes were made to the text. In addition, we clarified the region of the study in the Abstract: *"The index is computed for the full period available from the Iberian Gridded Dataset (1971 to 2015), focusing on the Iberian Peninsula."*.

2. Abstract: Some kind of numerical cuantification of how the proposed GDI improves previous indices could be indicated, as a summary of the discusion and conclusions section, although there not many numbers are shown, but if the authors can give some more specific quantification of such improvement, it could improve abstract paragraph. The fact that daily scales are used is also quite relevant, but it is just stated on the last phrase there. Perhaps it could be stated before and more clearly.

We thank the reviewer for the suggestions; in agreement, we performed some changes to the abstract in order to include some results quantification, and also to accommodate the changes proposed on the previous answer:

*"Drought is a complex climatic phenomenon characterised by water scarcity and is recognised as the most widespread and insidious natural hazard, posing significant challenges to ecosystems and human society. In this study, we propose a new daily-based index for characterising droughts, which involves standardising precipitation and/or precipitation minus potential evapotranspiration data. The new index, Generalized Drought Index (GDI), proposed here is computed for the full period available from the Iberian Gridded Dataset (1971 to 2015), focusing on the Iberian Peninsula region. Comparative assessments are conducted against the daily Standardized Precipitation Index (SPI), the Standardized Precipitation Evapotranspiration Index (SPEI), and a simple Z-Score standardisation of climatic variables. Seven different accumulation periods are considered (7, 15, 30, 90, 180, 360, and 720-days) with three drought levels: moderate, severe, and extreme. The evaluation focuses mainly on the direct comparison amongst indices, in their ability to conform to the standard normal distribution, added value assessment using the Distribution Added Value (DAV) and a simple bias difference for drought characteristics. Results reveal that the GDI together with the SPI and SPEI follow the standard normal distribution. In contrast, the Z-Score index depends on the*

*original distribution of the data. The daily time step of all indices allows the characterisation of flash droughts, with the GDI demonstrating added value when compared to SPI and SPEI for the shorter and longer accumulations, with positive DAV up to 35%. In comparison to the Z-Score, the GDI shows expected greater gains, particularly at lower accumulation periods, with DAV reaching 100 %. Furthermore, an assessment of the spatial extent of drought for the 2004-2005 event is performed. All three indices generally provide similar representations, except for the Z-Score, which exhibits limitations in capturing extreme drought events at lower accumulation periods. Overall, the findings suggest that the new index offers improved performance and adds value comparatively to similar indices with a daily time step."*

Regarding the daily scale index, we already mention and emphasise these scales in the second phrase.

3.  Introduction, lines 91-95: More clear statements about the differences between model and observed data should be made, as they are quite different, and there they seem to be two similar types of climate information. A comment that both are gridded data, for example, should be more clearly stated, and the need of having daily resolution. When talking about observational gridded data, perhaps a comment about other gridded daily precipitation databases could be made (E-OBS, GPCP or CPC ones)?.

As mentioned before, the EURO-CORDEX has been removed from the manuscript, however, in this context, the CORDEX datasets are still mentioned. We thank the reviewer for the suggestions, and we added a comment on other daily-based datasets:

*"Flash droughts can only be identified with daily drought indices, due to their sub-monthly timescale nature. Therefore, the widespread of observational-based daily gridded datasets such as the Iberian Gridded Dataset for the full Iberia Peninsula (IB01; Herrera et al., 2019),the Climate Prediction Center (CPC, Xie et al., 2007; Chen et al., 2008), the E-OBS (Cornes et al., 2018), the European Meteorological Observations (EMO-5, Thiemig et al., 2022), station based datasets such as the European Climate Assessment & Dataset (ECA&D, Klein-Tank et al., 2002), reanalysis data such as ERA5 (Hersbach et al., 2020; 2023), the JRA-55 (Kobayashi et al., 2015), the Merra-2 (Gelaro et al., 2017), or regional climate models initiatives such as the World Climate Research Program Coordinated Regional Climate Downscaling Experiment (Giorgi et al., 2009; 2021, Gutowski et al., 2016), assisted the development of new drought indices with a daily time step (Wang et al., 2015; 2021; 2022; Jia et al., 2018; Li et al., 2020; Ma et al., 2020; Onuşluel Gül et al., 2021; Zhang et al., 2022a; 2022b; Zhang et al., 2023)."*

4.  Introduction/objectives, around line 126: Why IB region is chosen, some arguments could be made to justify such region, which, of course is a good one to analyze drought features. Also state at that point which time period is proposed would be fine there, although it is detailed later on methods. Nevertheless, one of my main concerns, partly indicated before, is about how Regional climate modelling EuroCORDEX material is presented. No doubt about its importance and relevance to be used for climate studies such as the one presented here. But it is confusing to me the way they are employed. Here it is a paper about a new drought index proposal, which is welcomed and quite interesting, and not an evaluation of RCMs when dealing with this index proposal. The first part about how it compares with other indices when using observations is fine. So RCMs are used not to see how they represent each index, but to obtain a potential "added value'' of this index compared with the others in terms of how models compare with observations using the Perkins skill score. In that sense, is strange not to present how EuroCORDEX models describe each drought index. At least a more detailed description of how precipitation is described from them could be of interest, as just some very brief remarks are made around lines 205-210. I would be more clear and precise when talking about "evaluation'' RCM simulations (that

is, reanalysis or perfect boundary conditions forced simulations), to clarify that their usage is reasonable in the frame that is proposed here. In summary, an effort to clarify and differenciate observations and regional climate models interst and usage in the paper would be important for the study to be more precise and robust.

We thank the reviewer for these comments and suggestions. The Iberian Peninsula is a region greatly affected by intense droughts, while at the same time it features a myriad of different climates. From maritime in the northwest to Mediterranean and semi-arid in the southeast. To this end, we considered the Iberian Gridded dataset (IB01, Herrera et al., 2019) as a baseline for computing the GDI and all the other indices described to evaluate the performance of the proposed index. At the same time, we decided to remove the EURO-CORDEX from the results section, since not only it caused some confusion but also the paper is about the proposal of a new daily drought index. Due to removing the EURO-CORDEX, the full period of IB01 is now considered (1971-2015; i.e. 45 years). This dataset was built considering a large number of station data from Portugal and Spain, offering a higher quality gridded dataset for the Iberian Peninsula than the E-OBS, for instance. In the future, we have the intention of producing a paper with the GDI index covering the whole world and using reanalysis data and possibly data from CMIP6 as well.

The removal of the EURO-CORDEX data implies some differences in the figures and their main goal. First, the q-q plot (Fig. 2) and statistics shown in Figure S3 are used to assess if the GDI conforms to the standard normal distribution, while at the same time checking the behaviour of the SPI, SPEI and Z-Score. The now Table 2 is a version of the DAV metric in which the reference is now the standard normal distribution. Thus, there is added value if the GDI is closer to the standard normal distribution than the other indices. Figure 3 for the time series, Table 3 and Figure 5 for the drought characteristics are used to assess the differences that exist amongst the indices. The case study does not feature any differences in its content. Please check the revised document for all the changes. The numbering of figures and tables are referent to the revised manuscript.

Herrera, S., Cardoso, R. M., Soares, P. M., Espírito-Santo, F., Viterbo, P., and Gutiérrez, J. M.: IB01: A new gridded dataset of daily precipitation and temperatures over Iberia. Earth System Science Data, 11(4), 1947-1956, doi:10.5194/essd-11-1947-2019, 2019.

5. Data and methods. Line 151: Only Soares et al., 2023b is cited as a reference for future IB droughts, but some others could be added, such as Quintana-Segui et al, 2016: https://hal.science/hal-01401386/, for example. I guess there are several other studies that could be named here. Even more specific studies, such as Sanchez et al., 2011 (doi: 10.1007/s10584-011-0114-9), using dry spells index, or other similar ones, could fit here.

We thank the reviewer for the nice suggestions, and we added the mentioned papers and the following paper to the manuscript:

Moemken, J., Koerner, B., Ehmele, F., Feldmann, H. and Pinto, J.G.: Recurrence of drought events over Iberia. Part II: Future changes using regional climate projections. Tellus, 74, 262. doi:10.16993/tellusa.52, 2022.

6.    Data and methods, lines around 160s. The selected time period for the analysis is not totally clear there, as IB01 ranges from 1971 to 2015, 1989-2008 is the ERainterim-forced period for the 12 RCMs, but no clear statement is made until the end of the section, on line 389, and then on the final section, in line 607. Maybe it would be better to state it more clear and earlier on the text, even on the objectives paragraph, as pointed before.

The reviewer is right, and we understand their comment. Due to the changes made in the manuscript, now the period ranges from 1971-2015 and was corrected throughout the manuscript. We also added the period analysed in the Introduction section.

7.    When describing EuroCORDEX ensemble, please be careful with expressions such as "daily synchronized climate data" (line 190), which should be more clearly explained, as it is not clear there if they refer to reanalysis or GCM forced simulations, just mentioned before. In any case, that statement is not usual, even if it referred to reanalysis-forced simulations, so a more clear explanation should be made there.

This section has been removed from the manuscript, since now we do not consider the EURO-CORDEX dataset anymore.

8.    In equation 19, GTI should be GDI in the subindex?

Yes, corrected, we thank the reviewer for noticing this mistake.

9.    Related to previous comments about observations/models, a more precise description of some subsection titles could be considered. I mean: 3.1 GDI general performance "using gridded observations", or 3.3. GDI added value using RCM ensembles, or something like this?

Since the EURO-CORDEX was removed from the study, we decided to change the name of the sub-sections accordingly:

3.1. Remains as is: "GDI general performance"

3.2. Changes to "Drought Characteristics Assessment"

3.3 Is now the case study: "2004/2005 Case Study".

10.  My other main concern is about figures' representation or description, that could be clarified or better described, to my opinion. For example, I am not sure to fully understand colors in figure 2. Do they represent the same as the X axis, that is, the percentile?. It is therefore redundant?. And numbers on Y axis are probabilities of normal distribution?. Naming and describing properly axis on the figures and on the captions is essential for a good understanding of figures, together with the idea of being as simple and clear as possible using the minimum/optimum amount of information.

We thank the reviewer for the comment. We changed the figure by removing the colours and replacing each point with open circles and add to the legend and y-axis a mention to the "standard normal distribution percentiles (SND percentile)".

11. It happens to me again on figure 3: RMSE and time correlation (described in line 341) are obtained from the average of all the models, or for each model, and then the average of correlations in some other way?. This was my first thought, before realizing that all these section 3.1 was made only from IB01 gridded observational database. Please, make figures and section more clear with respect to the used information, for a reader to have clear which data is being used.

We removed the EURO-CORDEX data. Now this figure has changed to Figure 3. Nevertheless, this figure only considered the IB01 dataset land points. We also removed the colours. Now the sense of density is given by the superposition of the data, since each line is semi-transparent.

12. In that figure, and in general in others where all the cells/locations are included, all of them are considered as equal. I mean, there lines represent each IB point, but this approach misses some insight about subregions, coastal points, mountain areas that could add information about spatial features. Did the authors make a thought of maybe using rough subregions with homogeneous climatic conditions to performe at least some of the computations?. This idea also came to me when looking at boxplots for all the cells and models on figure 6. Besides, on that figure 3, density values are strange to me, in the sense that their units, or their maximum value, could be stated or indicated in some way, for a better understanding of such numbers?

As stated in the previous comment, the sense of density now is given by the tightly packed values to each other in Figure 3. At the beginning of this work, we started by making maps. However, for the results shown in Figures such as 2, 3 and Table 2, the values were very similar to each other and quite noisy, making it difficult to make a spatial evaluation. Besides, the main goal with these figures is to test for normality, check added value and compare the different indices. Of course, different regions may reveal different performances, which is given by the spread on specific figures.

13. Table 2: What blue and red colors mean?. How drought intensity, mean decadal frequency and mean duration are defined?.

Due to the changes in the order of the figures, now the numbering of this table changed to Table 3. The colours are used to highlight values. For instance, correlation values above 0.9 are marked in blue, correlation values below 0.7 are marked in red, while the other values are black. A colour bar is added to the table in order to the reader understand their meaning.

14. Table 3: I am not sure what "aggregation of the results obtained for all models'' mean. As the percentage of land points with a certain DAV, it means that each cell has 12 DAV values, one for each model, and so the percentage is obtained from those 12 combinations of model against observations score?. The sum of the numbers at each column equals 100% then?. A title on Y axis would also be nice to be shown, following my request to a better description of figures and tables.

Due to the removal of the EURO-Codex, the content of the now Table 2 has also changed. Regardless, the y-axis represents the DAV values. The sum for each column must be 100 % and corresponds to the total of land points, where each box represents the percentage of land points which has value within that category. For instance, on Table 2a for the PR-based index at the 7-Day accumulation, 52.062 % of land points have a DAV value between 10 to 15 %. To avoid confusion, we added/updated the captions to the y-axis and the colour bar.

15.  One final typographic remark, related to resolution numbers of gridded data: degrees should be shown as superscripts, but this is not made in lines 150-160

Corrected.

**Main Changes to the Manuscript:**

- Removal of the EURO-CORDEX dataset from the analysis and consequently removal of section 2.1.2
- Remake of all figures considering only the Iberian Gridded Dataset for the full period available (1971-2015, 45 years), instead of the original 20-years.
- Changes in text according to the change in the methodology.
- All figures were remade with this new period, where some figures (Fig 2 and 3 for example) were slightly simplified.
- Changes of sub-section numbering on sections 2 and 3:
    - (Section 2)
    - 2.1      Study Area
    - 2.2      IB01 Observational Dataset
    - 2.3      Potential Evapotranspiration
    - 2.4      Drought Indices
    - 2.4.1  Standardised      Precipitation      and      Standardised      Precipitation Evapotranspiration Indices
    - 2.4.2  Z-Score Index
    - 2.4.3  Generalized Drought Index
    - 2.5      GDI Evaluation
    - (Section 3)
    - 3.1      GDI General performance
    - 3.2      Drought Characteristics Assessment
    - 3.3      2004/2005 Case Study
- Section 2.5 main change is related to the DAV metric. The reference is now the standard normal distribution.
- Section 3. Change figures/table position:
    - Table 2 -> Table 3
    - Table 3 -> Table 2
    - Figure 6 -> Figure 4
    - Figure 4/5->Figure 5/6
- Change and correction to the text referring to Figure 2-4 and to Tables 2 and 3.
- Monthly analysis added to the supplementary material and figure/table numbering revised throughout the main manuscript.
- Small Corrections of typos and others across the entire manuscript.

---

## Author Response (AR2)

**Editor Comment:**

The authors have adequately addressed the comments of the reviewers. However, in doing so might have decreased the suitability of the paper for GMD. As the GMD webpage states, papers of interest to GMD include "new methods for assessment of models, including work on developing new metrics for assessing model performance and novel ways of comparing model results with observational data". The current manuscript certainly meets the journal's interest in "developing new metrics for assessing model performance". However, the first iteration of the paper was especially interesting as it went a step further and applied the work to Euro-CORDEX. I understand the reasons that the authors gave for removing it in this iteration of the manuscript; however, in doing so, it loses a clear connection to model evaluation. I suggest that the manuscript be amended to make it more clear throughout the paper the relevance of this work to model evaluation. As it currently stands, it only reads as a tool to take observational datasets and compute a drought index. How will this work help model evaluation (e.g., CMIP models)?, what does it bring to the table that was not there before? The authors already hint at this in the discussion/conclusions where they talk about the WCRP Flagship Pilot Study. I think taking these explaining these types of ideas further would help the paper. My suggestion is to edit/rewrite the introduction and discussion to make this connection more clear and succinct. Note that I am not asking for further analysis or figures, simply to address these sections to make the connection and applicability to climate models, weather models, etc... much more clear. This will benefit the paper as it will help bridge the gap from drought index development to model evaluation more clearly.

We wish to express our gratitude to the editor for the positive and constructive feedback on our study. We believe that the revisions have significantly improved the manuscript and strengthened its connection with model evaluation. The changes made have undoubtedly enhanced the overall quality of the manuscript. Regarding the changes, two sections were added, one in the introduction and another on the discussion Furthermore, some small changes and corrections were performed throughout the entire document.

[revised manuscript text omitted]